# Evaluation of Survival, Recurrence Patterns and Adjuvant Therapy in Surgically Staged High-Grade Endometrial Cancer with Retroperitoneal Metastases

**DOI:** 10.3390/cancers13092052

**Published:** 2021-04-23

**Authors:** Jennifer McEachron, Lila Marshall, Nancy Zhou, Van Tran, Margaux J. Kanis, Constantine Gorelick, Yi-Chun Lee

**Affiliations:** 1Division of Gynecologic Oncology, SUNY Downstate Medical Center, Brooklyn, NY 11203, USA; lila.marshall@downstate.edu (L.M.); nancy.zhou@downstate.edu (N.Z.); van.tran@downstate.edu (V.T.); yi-chun.lee@downstate.edu (Y.-C.L.); 2Division of Gynecologic Oncology, New York Presbyterian—Brooklyn Methodist Hospital, Brooklyn, NY 11203, USA; mak9273@nyp.org (M.J.K.); cog9013@nyp.org (C.G.)

**Keywords:** endometrial cancer, retroperitoneal metastasis, chemoradiation

## Abstract

**Simple Summary:**

High-grade endometrial carcinomas present a clinical challenge due to their propensity to metastasize beyond the pelvis. Stage IIIC disease represents a unique entity relative to all other stages, as the retroperitoneal lymph nodes are involved. Prior randomized controlled trials have suggested that chemotherapy alone is sufficient to treat stage III and IV endometrial carcinoma. However, no prospective trial has specifically evaluated adjuvant therapy strategies in stage IIIC disease; therefore, it is difficult to generalize the results of trials including all advanced stage patients to this unique group. In the present study, we observed improved progression-free and overall survival in those patients with stage IIIC disease receiving a combination of both chemotherapy and radiation, suggesting that radiation plays a key role in the management of retroperitoneal metastasis and should be considered in addition to chemotherapy.

**Abstract:**

Background: We seek to evaluate the difference in recurrence patterns and survival among stage IIIC high-grade endometrial cancer treated with surgery followed by adjuvant chemotherapy alone, radiation therapy alone, or both (chemoradiation). Methods: A multicenter retrospective analysis of surgically staged IIIC HGEC receiving adjuvant therapy was conducted. HGEC was defined as grade 3 endometrioid adenocarcinoma, serous, clear cell and carcinosarcoma. Differences in the frequency of recurrence sites and treatment delays were identified using Pearson’s χ^2^ test. Progression-free survival (PFS) and overall survival (OS) were calculated using Kaplan–Meier estimates. Results: A total of 155 patients were evaluable: 41.9% carcinosarcoma, 36.8% serous, 17.4% grade 3 and 3.9% clear cell. Of these, 67.1% received chemoradiation, 25.8% received chemotherapy and 7.1% received radiation therapy. There was no difference in the frequency of treatment delays between regimens (*p* = 0.571). There was a trend towards greater retroperitoneal recurrence with chemotherapy (25.9%) versus chemoradiation (8.4%) and radiation therapy (7.7%) (*p* = 0.252). Grade 3 tumors had improved progression-free and overall survival (26 and 42 months, respectively) versus serous (17 and 30 months, respectively), carcinosarcoma (14 and 24 months, respectively) and clear cell (24 and 30 months respectively) (*p* = 0.002, *p* < 0.001). Overall, chemoradiation was superior to chemotherapy and radiation therapy in PFS (*p* < 0.001) and OS (*p* < 0.001). Upon multivariate analysis, only histology and receipt of chemoradiation were independent predictors of survival. Conclusion: The majority of stage IIIC high-grade endometrial carcinomas recurred. Chemoradiation was associated with improved survival and less retroperitoneal recurrence. Grade 3 tumors demonstrated improved survival versus other histologies regardless of adjuvant treatment modality.

## 1. Introduction

Endometrial cancer is the most common gynecologic malignancy. Since the early 2000s, its incidence has been on the rise with a 1% increase in number of cases per year. Based on estimates from the National Cancer Institute, there will be 65,620 new cases by 2020 [1]. Its incidence is increasing particularly in the developed world, in part due to an aging population; however, the increase in obesity and metabolic syndrome in these regions is a key contributing factor [2,3]. The majority of endometrial cancer is diagnosed at an early stage and associated with excellent 5-year survival, exceeding 90% for stage I disease. However, over one-quarter of patients will present with disease beyond the pelvis [2,3,4,5]. Although excellent outcomes are observed in early-stage disease, survival declines dramatically with advancing stages at diagnosis [6,7,8].

Stage III EC represents a broad array of metastatic spread patterns, limiting the applicability of generalizing “stage III disease” as one specific entity. Stage IIIA and IIIB represent metastatic spread to pelvic structures, vastly different than stage IIIC disease, which involves spread to retroperitoneal lymph nodes. Formerly considered together as stage IIIC disease, current FIGO staging defines endometrial carcinoma involving the pelvic lymph nodes as stage IIIC1, and cancer involving the para-aortic lymph nodes as stage IIIC2. This update reflects the appreciation of retroperitoneal lymph node involvement, particularly the para-aortic lymph nodes, as a poor prognostic indicator [8]. The 5-year survival declines from 60 to 70% with metastasis to the pelvic lymph nodes, dropping rather dramatically to 30–40% with para-aortic lymph node involvement [2,8].

Traditionally, endometrial cancer has been classified into two main categories. Type I tumors, classically described as low-grade endometrioid adenocarcinoma, are responsible for 80% of all EC. These tumors are estrogen-driven and tend to present at an early stage, leading to an overall good prognosis [9]. In contrast, type II tumors are much more likely to present at advanced stages, harbor a high recurrence rate and, subsequently, a significantly poorer survival. Type II tumors traditionally include serous, clear cell, carcinosarcoma, and undifferentiated carcinoma. Although these tumors only represent 15–20% of all EC cases, they account for over 50% of all deaths due to their propensity for distant metastasis and chemoresistance [2,10].

More recently, The Cancer Genome Atlas classified endometrial carcinoma into four categories based on molecular characteristics: POLE ultramutated, microsatellite instability hypermutated, copy-number low and copy-number high [11]. The first three groups are most often associated with low-grade endometrioid cancers and have favorable or intermediately favorable prognoses. This is in contrast to the copy-number high group, or “serous-like” group, which is associated with non-endometrioid histology, advanced stage at diagnosis, and p53 mutations. Interestingly, The Cancer Genome Atlas investigators found that 25% of grade 3 endometrioid tumors possess a molecular phenotype similar to uterine serous carcinomas, suggesting the clinical behavior of these high-grade endometrioid tumors resembles that of serous carcinomas rather than their low-grade endometrioid counterparts. Based on these findings, uterine serous carcinoma, clear cell carcinoma, carcinosarcoma and grade 3 endometrioid adenocarcinoma are classified as “high-grade endometrial carcinomas”, uniquely different in clinical outcome compared to grade 1 and 2 endometrioid carcinomas.

Adjuvant therapy for advanced-stage disease is centered around systemic chemotherapy [4]. Randomized clinical trials have consistently demonstrated a survival benefit with the use of chemotherapy in extrauterine disease [5,6,7]. However, chemotherapy alone has been associated with an increase in local recurrence, with pelvic relapse rates ranging from 18 to 40% [12,13,14]. Despite this, the role of radiation therapy in this patient population is less well-defined [2,13,15,16,17,18]. Multiple authors have reported that combination adjuvant therapy with both chemotherapy and radiotherapy yields superior clinical outcomes compared to either modality alone [16,17,18,19]. However, there is conflicting prospective data to support a survival benefit with the addition of radiation to systemic therapy [13,19]. Importantly, the majority of studies evaluating adjuvant therapy often generalize advanced-stage disease as all stage III and IV disease, not accounting for the critical involvement of retroperitoneum observed in stage IIIC disease. In the present study, we review our experience with stage IIIC high-grade endometrial carcinoma to evaluate clinicopathologic factors and determine the optimal adjuvant therapy strategy for this patient population.

## 2. Methods

From 2000 to 2019, a multicenter retrospective analysis of patients with stage IIIC high-grade endometrial carcinoma was conducted. Internal review board approval was obtained at all participating sites. Tumor registries were reviewed to identify all patients with stage IIIC high-grade endometrial carcinoma who received primary surgical treatment, followed by adjuvant therapy with chemotherapy alone, radiation therapy alone, or a combination of chemoradiation. High-grade tumors were defined as grade 3, serous, clear cell and carcinosarcoma. Primary surgical management was defined as hysterectomy, with or without bilateral salpingoophorectomy, and comprehensive surgical staging consisting of pelvic + para-aortic lymph node dissection +/− omentectomy. Chemoradiation was defined as receipt of systemic chemotherapy, followed by, after or in sandwich sequence, external beam radiation therapy +/− vaginal brachytherapy. When available, information on mismatch repair (MMR) status/microsatellite instability (MSI) was collected. Tumors were classified as MMR deficient by absence of MMR proteins on immunohistochemistry or microsatellite instability identified on next-generation sequencing. Key exclusion criteria included histologic diagnosis of low-grade endometrioid or sarcoma, patients with incomplete surgical staging data and patients receiving preoperative pelvic radiation therapy and/or neoadjuvant chemotherapy.

Clinical and demographic data were obtained from tumor registries and a review of both inpatient and outpatient medical records. Extracted data included date of diagnosis, surgical procedures, types of adjuvant therapy, date and site of recurrent, chemotherapy regimen, number of chemotherapy cycle received, type of radiation therapy received and date of death. A one-way ANOVA test was used to compare differences in mean age between treatment arms. Differences in the frequencies of treatment delays and sites of disease recurrence were identified using Pearson’s chi-square test. Progression-free survival was defined as the time of initial surgical management to the time of first recurrence. Overall survival was defined as the time of initial surgical management to the time of death. Patients who were alive at date of last follow-up were censored. Progression-free and overall survival rates were calculated using Kaplan–Meier estimates and the log-rank test. The multivariate analysis was performed using the Cox proportional hazards model. Statistical significance was defined as *p* < 0.05. The analysis was performed using SPSS version 26.0 (IBM, Armonk, NY, USA).

## 3. Results

### 3.1. Patient Characteristics

From 2000 to 2019, there was a total of 190 patients identified with stage III high-grade endometrial carcinoma. The final analysis included 155 patients with stage IIIC1 and IIIC2 disease undergoing primary surgical management followed by adjuvant therapy. The mean age was 64 years (range 49–83), and the majority of patients were African-American (82%). Histology included 65 (41.9%) carcinosarcoma, 57 (36.8%) serous, 27 (17.4%) grade 3 and 6 (3.9%) clear cell. Stage distribution included 63 (41%) stage IIIC1 and 92 (59%) stage IIIC2. The majority of patients (74%) had lymphovascular space invasion. Twenty-eight (18.1%) patients had isolated positive para-aortic lymph nodes in the presence of negative pelvic lymph nodes. This included 22% of grade 3, 18% of carcinosarcoma, 16% of serous and 17% of clear cell tumors. All patients received adjuvant therapy, 104 (67.1%) received chemoradiation, 40 (25.8%) received chemotherapy and 11 (7.1%) received radiation therapy. Ninety-two patients (59.4%) had information on MMR status available for review. We found 12.3% of all patients were MMR deficient, with grade 3 tumors demonstrating the highest incidence of MMR deficiency relative to all other histologies (*p* = 0.005) (Table 1).

### 3.2. Adjuvant Therapy

The majority of patients (92.9%) received chemotherapy with or without radiation. Of these patients, 95.8% were treated with platinum-based chemotherapy, of which the most common regimen was carboplatin-paclitaxel (72.2%). Other treatment regimens included cisplatin-paclitaxel-doxorubicin (10.4%), cisplatin-ifosfamide (10.4%), paclitaxel-ifosfamide (2.8%), cisplatin-doxorubicin (2.1%), single-agent doxorubicin (1.4%) and carboplatin-paclitaxel-bevacizumab (0.7%). Treatment regimens did not differ significantly between chemotherapy alone and chemoradiation cohorts (*p* = 0.981). When excluding carcinosarcoma, there was no significant difference in treatment regimens between different histologic subtypes (*p* = 0.351). Although the most common regimen administered to patients with carcinosarcoma was carboplatin-paclitaxel (64.3%), a notable number of patients received ifosfamide-containing regimens (33.9%). The median number of cycles received was 6 (range 3–8). Of the patients receiving radiation therapy, the majority received external beam radiotherapy to the pelvis with extended para-aortic field plus vaginal brachytherapy (54.6%). The remaining received a combination of EBRT +/− extended field without brachytherapy (45.4%).

### 3.3. Recurrence Patterns

A total of 128 patients recurred during the study period, with 162 individual sites of disease recurrence. Although the incidence of recurrence did not differ significantly based on tumor histology, grade 3 tumors recurred less frequently (66.7%) compared to serous (80.7%), clear cell (83.3%) and carcinosarcoma (84.6%) (*p* = 0.269). The most common location of disease recurrence was the abdomen (45.7%), followed by the pelvis (14.2%), retroperitoneum (14.2%), vagina (4.3%) and extra-peritoneal distant sites (1.2%). The distribution of recurrence sites did not differ significantly between histologic subtypes (*p* = 0.8934). However, we observed a trend towards more abdominal recurrence in clear cell (67.0%), serous (50.0%) and carcinosarcoma (42.6%) compared to grade 3 endometrioid (39.2%) histology. Carcinosarcoma was the most likely histology to recur in the lung (19.1%) and at extra-peritoneal sites (3.0%). The distribution of recurrence did not differ significantly between adjuvant therapy strategies (*p* = 0.252). However, there was a trend towards greater retroperitoneal recurrence with chemotherapy (25.9%) compared to chemoradiation (8.4%) and radiation (7.7%).

### 3.4. Treatment Outcomes

The median follow-up was 5.5 years. The median progression-free survival of the entire cohort was 16 months, and the median overall survival was 30 months. The median progression-free survival was significantly improved in grade 3 (26 months) compared with serous (17 months), carcinosarcoma (14 months) and clear cell (24 months) tumors (*p* = 0.002). The median overall survival was also significantly longer in patients with grade 3 (42 months) versus serous (30 months), carcinosarcoma (24 months) and clear cell (30 months) tumors (*p* < 0.001) (Figure 1).

Across all histologies, the median progression-free survival was improved with chemoradiation compared to chemotherapy and radiation (18 months vs. 12 months and 8 months, respectively) (*p* < 0.001). Similarly, the median overall survival was also significantly improved across the entire cohort with a combination of chemoradiation versus chemotherapy and radiation (35 months vs. 22 months and 13 months, respectively) (*p* < 0.001) (Figure 2). Grade 3 tumors experienced the largest progression-free and overall survival gain with combination therapy, which improved progression-free survival by 14 months and overall survival by 24 months, versus chemotherapy alone. Serous and carcinosarcoma histology also demonstrated clinically significant gains in progression-free (6 and 5 months, respectively) and overall survival (8 and 9 months, respectively) (Table 2) (Appendix A). According to the multivariate analysis, only histology and receipt of chemoradiation were independent predictors of survival. Age, the presence of lymphovascular space invasion and stage (IIIC1 vs. IIIC2) did not independently alter survival outcomes (Table 3).

## 4. Discussion

Although chemotherapy and radiation, either as single modalities or in combination, are often used in the treatment of stage IIIC EC, the optimal adjuvant therapy strategy remains controversial. In the current study, we observed a significant improvement in both progression-free and overall survival among patients treated with combination chemoradiation versus chemotherapy or radiation therapy alone, for each histology individually and as an entire cohort. Our findings align with prior retrospective analyses conducted by Secord et al. and Albeesh et al., which both observed improved progression-free and overall survival among patients with advanced endometrial carcinoma receiving a combination chemoradiation versus either modality alone [17,18]. However, our observations contrast with the recently published randomized phase III trial conducted by the Gynecologic Oncology Group (GOG) 258, which found that combination adjuvant chemoradiation did not significantly improve recurrence-free survival compared to chemotherapy alone [13].

A number of important differences exist between the current study and GOG 258, which likely contribute to the difference in outcomes. The sequence of chemoradiation administered in GOG 258 was radiation followed by chemotherapy. Although we did not perform a separate sub-analysis based on the sequence of chemoradiation in our patients due to insufficient numbers, the majority received chemoradiation followed by chemotherapy prior to radiotherapy. Additionally, over 50% of the patient population of GOG 258 was composed of grade 1 and 2 endometrioid adenocarcinoma, in contrast to the current report, which only comprised high-grade tumors. Importantly, our findings do correspond with the recently published PORTEC-III, which found improved 5-year failure-free and overall survival benefit with combination chemoradiation versus radiation therapy alone [19]. The authors believe this comparison to be more relevant than that of the GOG 258, based on the results of the pre-planned sub-analysis of stage III patients demonstrating significant survival benefits from the combination of chemoradiation.

We observed an increase in retroperitoneal recurrence in patients receiving adjuvant chemotherapy alone versus chemoradiation. These findings agree with existing retrospective as well as prospective data, suggesting the importance of radiation therapy in retroperitoneal disease control [13,20,21,22]. Additionally, we observed a higher incidence of isolated positive para-aortic lymph nodes (18.1%) than previously reported in the literature. Abu-Rustum et al. and Chiang et al. reported an isolated para-aortic lymph node metastasis rate of only 1.2–1.6% [23,24]. Importantly, these reports included surgically staged stage I–IV disease, contributing to the lower incidence. However, Abu-Rustum and colleagues failed to report the histologic composition of the study population, and Chiang’s patient population was composed of 95% endometrioid histology, of which 88% were grade 1 and 2 disease. The incidence of isolated para-aortic lymph node involvement is reportedly higher in type II carcinomas; Goff and colleagues reported an incidence of 12% in uterine serous carcinoma [25]. Isolated positive para-aortic lymph nodes have been associated with the presence of lymphovascular space invasion, and 74% of the tumors in our study displayed lymphovascular space invasion, which could have contributed to the high incidence of stage IIIC2 disease and isolated positive para-aortic lymph node metastasis [26]. This finding emphasizes the importance of para-aortic nodal assessment during surgical staging.

Parallel with previous reports, the present study observed an improvement in both progression-free and overall survival among endometrioid histology when compared to all other histologies, regardless of the type of adjuvant therapy [27,28,29]. Notably, we found endometrioid carcinomas derived a significant survival benefit with the combination of chemoradiation compared to chemotherapy alone. Unfortunately, we could not compare chemoradiation versus radiation alone in our population, as the radiation alone cohort was composed of only one patient. The PORTEC III demonstrated improved survival and decreased recurrence with the combination of chemoradiation in all histology, which further supports the adjuvant therapy using chemoradiation over single modality therapy [19]. However, a post hoc analysis of the PORTEC III, based on molecular tumor classification rather than histologic subtype, demonstrated improved survival with the combination of chemoradiation only in those tumors expressing p53. Notably, 23% of tumors were found to express p53, and only 16% were histologically classified as serous. The remaining 7% of p53-expressing tumors were endometrioid, suggesting the importance of molecular classification to guide adjuvant therapy, rather than histologic classification [30]. Additionally, recent literature has investigated other genes as prognostic indicators irrespective of histology. Specifically, ARID1A and HER2/neu mutations have been identified as negative prognostic indicators across all histologic subtypes [31,32], which again illustrates that the molecular classification of tumors is critical when considering adjuvant therapy.

Although the current study, and a wealth of retrospective literature, supports the use of chemoradiation, the optimal sequence of adjuvant chemoradiation remains unclear. Concerns with receipt of radiation prior to chemotherapy include opportunistic distant metastasis in the absence of systemic therapy, and the obliteration of the tumor vascular bed by radiotherapy, potentially impairing future chemotherapy delivery and efficacy [17,33]. Additionally, theories suggest that if chemotherapy is given before radiation, its toxicity may result in radiation delays and interruption of radiotherapy [34]. Goodman et al. studied stages III–IVA EC and found chemotherapy followed by radiation therapy was superior to both the alternate sequence and either modality alone [16]. Secord and colleagues evaluated sandwich sequencing in advanced endometrial cancer [17]. Sandwich sequencing is defined as chemotherapy administered both before and after radiation therapy. These authors observed a significant improvement in both progression-free and overall survival with the sandwich sequence compared to other sequential treatment sequences. Similarly, Lu et al. observed a significant improvement in survival with sandwich sequencing in a cohort of stage IIIC endometrioid EC versus alternate chemoradiation sequencing [35]. Taken together, these studies provide a clear indication for continued exploration of the optimal sequence of adjuvant therapy modalities.

As the current report is retrospective, it is inherently limited by a degree of selection bias. Additionally, due to the 19-year time period included, there have been changes in the use of specific chemotherapy regimens over time. Despite these limitations, the majority of patients received platinum-based chemotherapy, which currently remains the preferred front-line therapy for advanced endometrial cancer when systemic therapy is selected. Several key strengths of this study included a long follow-up period and a large sample size relative to similar prior reports. Despite these limitations, we observed a significant improvement in both progression-free and overall survival with chemoradiation in a high-risk patient population, specifically with retroperitoneal nodal metastases.

## 5. Conclusions

The majority of stage IIIC high-grade endometrial carcinomas will recur. Grade 3 endometrioid carcinomas carry a more favorable prognosis relative to all other high-grade carcinomas regardless of adjuvant treatment modality. Importantly, we found 18% of patients had isolated positive para-aortic lymph nodes, illustrating the importance of surgical evaluation of the para-aortic region to ensure adequate adjuvant therapy. The combination of chemoradiation offered improved progression-free and overall survival in this unique patient population, attributed to the decrease in retroperitoneal compared to adjuvant chemotherapy alone. The 13-month and 22-month improvements in overall survival achieved with chemoradiation versus chemotherapy alone and radiation therapy alone, respectively, are clinically significant observations that warrant further prospective evaluation.

## Figures and Tables

**Figure 1 cancers-13-02052-f001:**
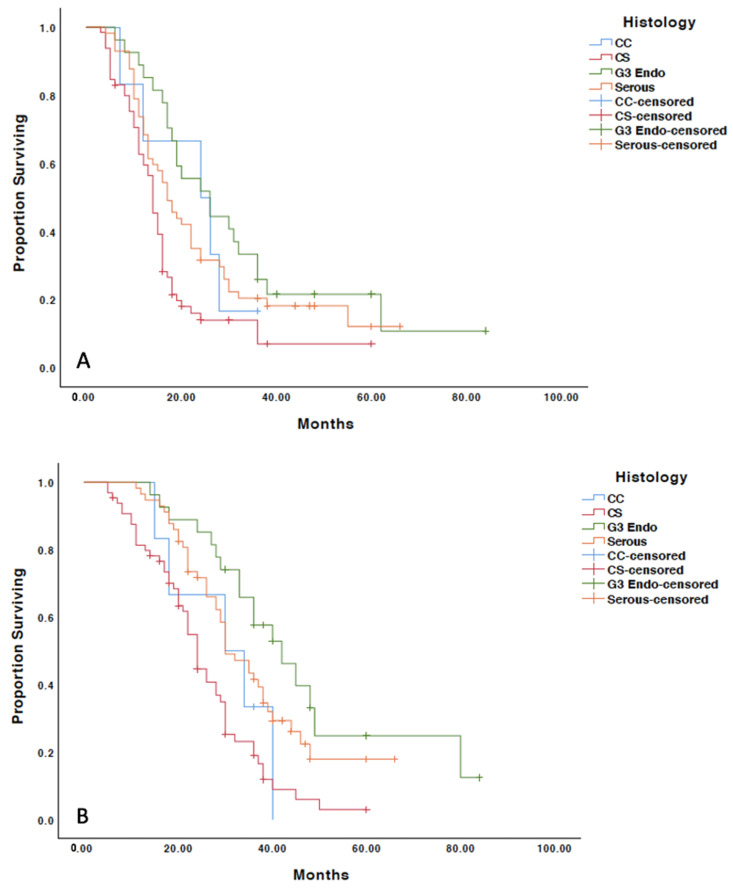
Kaplan–Meier survival analysis by histology. (**A**) Progression-free survival analysis; (**B**) overall survival analysis; CC: clear cell; CS: carcinosarcoma; G3 Endo: grade 3 endometrioid.

**Figure 2 cancers-13-02052-f002:**
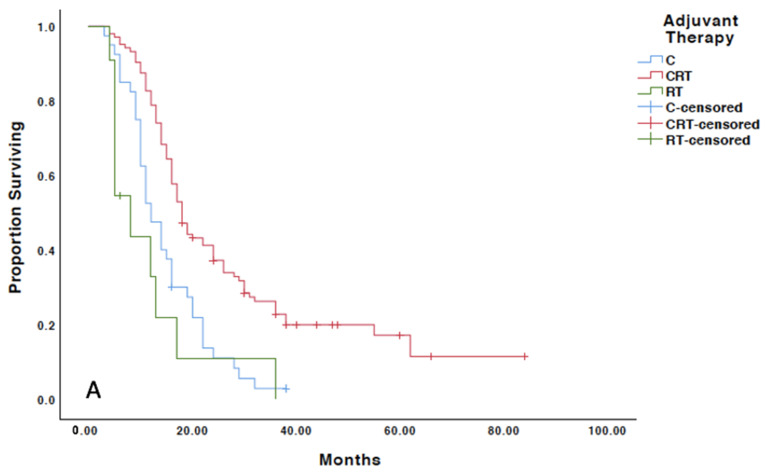
Kaplan–Meier survival analysis based on adjuvant treatment regimen. (**A**) Progression free survival analysis; (**B**) overall survival analysis; CT: chemotherapy alone; CRT: chemoradiation; RT: radiation therapy.

**Table 1 cancers-13-02052-t001:** Clinical and pathologic characteristics.

Characteristics	All Patients (*n* = 155)	Grade 3 Endometrioid (*n* = 27)	Serous (*n* = 57)	Carcinosarcoma (*n* = 65)	Clear Cell (*n* = 6)	*p*-Value
Age at surgery						0.273
Mean (range)	66 (43–85)	64 (49–80)	66 (49–81)	67 (43–85)	71 (56–83)	
	*n* (%)	*n* (%)	*n* (%)	*n* (%)	*n* (%)	
Race						0.463
Caucasian	25 (16)	5 (19)	6 (11)	13 (20)	1 (17)	
African-American	127 (82)	21 (78)	51 (89)	50 (77)	5 (83)	
Other	3 (2)	1 (3)	0 (0)	2 (3)	0 (0)	
FIGO Stage *						0.318
IIIC1	63 (41)	13 (48)	18 (32)	30 (46)	2 (33)	
IIIC2	92 (59)	14 (52)	39 (68)	35 (54)	4 (67)	
Presence of LVSI						0.434
Positive	114 (74)	17 (63)	41 (72)	51 (78)	5 (83)	
Negative	41(26)	10 (37)	16 (28)	14 (22)	1 (17)	
Adjuvant						0.043
therapy	104 (67)	23 (85)	41 (72)	36 (55)	4 (67)	
CRT	40 (26)	3 (11)	16 (28)	20 (30)	1 (16)	
Chemotherapy Radiation	11 (7)	1 (4)	0 (0)	9 (15)	1 (16)	
Sequence of CRT						0.193
CR	38 (37)	4 (17)	19 (46)	13 (36)	2 (50)	
CRC	36 (35)	10 (44)	15 (37)	10 (28)	1 (25)	
RC	30 (28)	9 (39)	7 (17)	13 (36)	1 (25)	
MMR Status						0.005
Deficient	19 (12)	9 (33)	6 (10)	3 (5)	1 (17)	
Unknown	63 (41)	6 (22)	21 (37)	34 (52)	2 (33)	
Proficient	73 (47)	12 (45)	30 (53)	28 (43)	3 (50)	

CRT: chemotherapy and radiation therapy; CR: chemotherapy followed by radiation therapy; CRC: chemotherapy-radiation-chemotherapy; RC: radiation therapy followed by chemotherapy. * 2009 FIGO Staging system.

**Table 2 cancers-13-02052-t002:** Progression-free and overall survival based on histology and adjuvant therapy.

Patient Characteristics	PFS	OS
All histologies		
All patients	16	30
Chemotherapy	12	22
CRT	18	35
RT	8	13
*p*-value	<0.001	<0.001
G3 Endometrioid		
All patients	26	42
Chemotherapy	12	18
CRT	26	42
*p*-value	0.027	0.003
Serous		
All patients	17	30
Chemotherapy	12	29
CRT	18	37
*p*-value	0.022	0.004
Carcinosarcoma		
All patients	14	24
Chemotherapy	11	21
CRT	16	28
RT	5	11
*p*-value	0.001	<0.001

CRT: chemotherapy + external beam radiation therapy +/− vaginal brachytherapy; OS: overall survival; PFS: progression-free survival; RT: radiation therapy. Statistical significance was defined as *p* < 0.05.

**Table 3 cancers-13-02052-t003:** Multivariate analysis for PFS and OS.

Variable	PFS			OS		
	HR	95% CI	*p*-Value	HR	95% CI	*p*-Value
Age (per year)	0.99	0.97–1.02	0.419	1.00	0.98–1.03	0.915
Presence of LVSI	0.69	0.47–1.02	0.06	0.71	0.47–1.09	0.115
Stage (IIIC1 vs. IIIC2)	0.79	0.55–1.61	0.242	0.86	0.57–1.28	0.451
Histology			0.012			0.019
Serous vs. G3	0.73	0.43–1.25	0.258	0.66	0.35–1.19	0.161
Serous vs. CS	1.62	1.08–2.45	0.019	1.64	1.04–2.57	0.032
Adjuvant therapy			0.001			<0.001
Chemotherapy vs. CRT	0.55	0.36–0.83	0.004	0.45	0.28–0.68	<0.001
Chemotherapy vs. RT	1.65	0.74–3.56	0.244	1.94	0.85–4.44	0.115

CRT: chemotherapy + external beam radiation therapy +/− vaginal brachytherapy; CS: carcinosarcoma; G3: grade 3 endometrioid; OS: overall survival; PFS: progression-free survival; RT: Radiation therapy. Statistical significance was defined as *p* < 0.05.

## Data Availability

Data are available upon reasonable request.

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
