# Peer review of "Evaluation of Survival, Recurrence Patterns and Adjuvant Therapy in Surgically Staged High-Grade Endometrial Cancer with Retroperitoneal Metastases"

_cancers, 2021, doi:10.3390/cancers13092052_

Round 1
Reviewer 1 Report
The manuscript is well prepared and the data presented here are interesting. However, the results are merely confirmativ and may be better suitable to a more specialized journal.
Minor comments:
In table 1 the abbreviation LVSI seems to be without explanation in the text body.
In fig. 1 the inset legend says CS, whereas the figure legend uses CSA.
Generally the tables should be revised with regard to line consistency. They are sometimes hard to follow in my copy.
Author Response
Reviewer 1:
- Thank you for your comments and input
- Table 1 legend has been revised to remove the abbreviation for LVSI
- Figure 1 legend has been revised to correct the abbreviation of carcinosarcoma
- The table legends have been revised to improve the consistency of abbreviations throughout
Reviewer 2 Report
The present work evaluates the difference in recurrence patterns and survival between high-grade stage IIIC endometrial cancer treated with surgery followed by adjuvant chemotherapy alone, radiation therapy alone, or both (chemoradiotherapy). The results show a trend toward greater retroperitoneal recurrence with chemotherapy compared with chemoradiotherapy and radiation therapy. Grade 3 tumors had better progression-free and overall survival than the other histotypes. The authors conclude that chemoradiotherapy is associated with better survival and less retroperitoneal recurrence. The work may be of interest and overall is well structured. Being a retrospective series I suppose that stratification based on surrogate TCGA classification (PORTEC/ProMiSe) cannot be considered. I consider this point unfortunately a major limitation in light of new evidences. In particular I wonder: what is the status of microsatellite instability in the studied cases? It is likely that endometrioid carcinomas may have a deficit of MMR and consequently have a different prognosis for this important feature. Can we assume that serous carcinomas and carcinosarcomas had TP53 mutations? Although it may not be easy for the authors to find these data, nowadays it would be important to know these alterations for an appropriate prognostic stratification. Moreover, among the high-grade histotypes, any undifferentiated/dedifferentiated carcinomas that should represent the same incidence as carcinosarcomas are not reported.
Author Response
Reviewer 2:
- Thank you for your comments
- MMR status: we have added information on the MMR/MSI status of all patients included to table 1. Unfortunately, there is a good number of patients for which we do not have this information available (which have described these patients has “unknown status”), however, based on those patients with available MMR data we observed a higher incidence of MMR deficiency in endometrioid tumors, as expected. The reason we did not include this data in the original manuscript draft was due to the amount of patients with missing MMR status. We acknowledge that it is relative to current treatment strategies.
- Regarding TP53 mutations: unfortunately, we do not have sufficient data evaluable to include this in our analysis. Although it cannot be assumed, we suspect a large portion of serous and carcinosarcomas do in fact harbor TP53 mutations – Of note; our team reached out to the authors of the GOG 258 regarding molecular stratification of these patients. They informed us that MMR and TP53 status were not collected in the GOG 258 patient cohort. Therefore, although this information would be interesting we feel it does not preclude publication.
- Our cohort does not include any patients with undifferentiated and dedifferentiated histology. Although we did have 5 such patients in our tumor registry, all 5 had major gaps in follow up data and were therefore unevaluable. The methods section has been modified to clarify that this specific cohort includes only grade 3 endometrioid, serous, clear cell and carcinosarcoma.
Round 2
Reviewer 1 Report
Thanks for your changes.
Author Response
No additional changes recommended. Thank you for your review.
Reviewer 2 Report
The authors responded to the reviewers' comments. I would emphasize in the discussion the importance of supplementing histologic data with markers that can identify cases with worse prognosis.
In particular, I strongly suggest citing some recent articles (also published in Cancers) important in the field that may also be of interest for eventual prognostic stratification and target therapy in endometrial cancer:
- Vermij L et al. HER2 Status in High-Risk Endometrial Cancers (PORTEC-3): Relationship with Histotype, Molecular Classification, and Clinical Outcomes. Cancers (Basel). 2020 Dec 25;13(1):44. doi: 10.3390/cancers13010044. PMID: 33375706; PMCID: PMC7795222.
- De Leo A et al. ARID1Aand CTNNB1/β-Catenin Molecular Status Affects the Clinicopathologic Features and Prognosis of Endometrial Carcinoma: Implications for an Improved Surrogate Molecular Classification. Cancers (Basel). 2021 Feb 25;13(5):950. doi: 10.3390/cancers13050950. PMID: 33668727; PMCID: PMC7956405.
- Joehlin-Price A et al. Molecularly Classified Uterine FIGO Grade 3 Endometrioid Carcinomas Show Distinctive Clinical Outcomes But Overlapping Morphologic Features. Am J Surg Pathol. 2021 Mar 1;45(3):421-429. doi: 10.1097/PAS.0000000000001598. PMID: 33021522.
- Asano H et al. L1CAM Predicts Adverse Outcomes in Patients with Endometrial Cancer Undergoing Full Lymphadenectomy and Adjuvant Chemotherapy. Ann Surg Oncol. 2020 Jul;27(7):2159-2168. doi: 10.1245/s10434-019-08103-2. Epub 2019 Dec 2. PMID: 31792716.
- Dondi G et al. An Analysis of Clinical, Surgical, Pathological and Molecular Characteristics of Endometrial Cancer According to Mismatch Repair Status. A Multidisciplinary Approach. Int J Mol Sci. 2020 Sep 29;21(19):7188. doi: 10.3390/ijms21197188. PMID: 33003368; PMCID: PMC7582893.
Author Response
Thank you for your comments and review.
- The authors have modified the discussion section to include a detailed discussion of the importance of molecular classification of tumors when considering adjuvant therapy.
- We have sited 2 of the recommended articles and study by Leon-Castillo et al. Molecular classification of the PORTEC III.
- Page 9 - Lines 285-295 (Additionally, the references section has been updated accordingly)